# Initiation and Prevention of Biological Damage by Radiation-Generated Protein Radicals

**DOI:** 10.3390/ijms23010396

**Published:** 2021-12-30

**Authors:** Janusz M. Gebicki, Thomas Nauser

**Affiliations:** 1Department of Biological Sciences, Macquarie University, Sydney 2109, Australia; 2Departement für Chemie und Angewandte Biowissenschaften, Eidgenössische Technische Hochschule, 8093 Zurich, Switzerland; nauser@inorg.chem.ethz.ch

**Keywords:** radiations, proteins, carbon radicals, antioxidants, radical adducts, kinetics

## Abstract

Ionizing radiations cause chemical damage to proteins. In aerobic aqueous solutions, the damage is commonly mediated by the hydroxyl free radicals generated from water, resulting in formation of protein radicals. Protein damage is especially significant in biological systems, because proteins are the most abundant targets of the radiation-generated radicals, the hydroxyl radical-protein reaction is fast, and the damage usually results in loss of their biological function. Under physiological conditions, proteins are initially oxidized to carbon-centered radicals, which can propagate the damage to other molecules. The most effective endogenous antioxidants, ascorbate, GSH, and urate, are unable to prevent all of the damage under the common condition of oxidative stress. In a promising development, recent work demonstrates the potential of polyphenols, their metabolites, and other aromatic compounds to repair protein radicals by the fast formation of less damaging radical adducts, thus potentially preventing the formation of a cascade of new reactive species.

## 1. Scope of the Review

Studies of the chemistry of actions of ionizing radiations on protein-rich materials range from the irradiation of protein crystals to living organisms. A basic observation shared by all studies is that invariably, the chemistry of the irradiated object is altered, with the alteration normally classified as a form of damage. The damage may lead to a useful outcome, as in radiation sterilization, but in other applications, it may need to be prevented.

While the subject of radiation effects has acquired increasing importance in basic chemistry and in industrial and defense applications, there is little doubt that the most widely pursued interests are in the field of medical applications of radiations in both diagnosis, such as X-ray imaging, and in treatment, most notably that of cancer. As usual in all such applications, a balance needs to be achieved between the benefit of the procedure and any damage, with the estimate of likely deleterious changes derived from the basic studies of radiation chemistry and biology [1,2,3].

This review focusses on the significance of proteins in absorbing and transferring the radiation-induced oxidative damage to other biomolecules in cells and tissues, and on tests of the ability of antioxidant compounds to repair the molecular damage sustained. For reasons of relevance and brevity, discussion is limited to particular conditions, corresponding to physiologically plausible states: an aqueous environment, near neutral pH, a complex organic medium, and the presence of oxygen.

## 2. Ionizing Radiations Generate Radicals in Aqueous Solutions

Here we consider only the actions of high-energy photons in X-ray and gamma rays and of accelerated electrons, leaving out high-energy nuclear particles and ultraviolet photons. By definition, the energy of the electrons and rays must exceed the binding energy of an electron in the irradiated material, resulting in the formation of ions, most commonly by the Compton effect [1]. In dilute aqueous solutions, virtually all of the radiation energy is absorbed by the water, present at about 50 M concentration. In the process, water is decomposed, and a range of well-characterized charged and neutral primary species are generated in known amounts. Some of them are reactive; under conditions relevant to biology, their radiation yields, measured in G values (units 10^−7^ mol J^−1^), are 2.8 hydroxyl (HO^●^) free radicals and hydrated electrons (e_aq_^−^), 0.55 H atoms, and 0.7 H_2_O_2_ [2]. In turn, the reactive primary intermediates can attack other solutes, generating secondary free radicals by electron transfer, initiating further reactions. Therefore, the radiation chemistry of aqueous solutions is in essence the chemistry of free radicals [3].

A major practical advantage of the application of radiation chemistry is the ability to alter the amounts and nature of the reactive intermediates by changing the quantity of the absorbed energy and by selective scavenging the primary radicals. The HO^●^ is commonly scavenged by *tert-*butanol, while the e^−^_aq_ can be converted to HO^●^ by saturating the solution with N_2_O. In addition, new radicals can be produced with diminished reactivity and greater selectivity, when, for example, the HO^●^ reacts with solutes such as formate, azide, or bromide, while the reduction of many compounds by the e_aq_^−^ produces a variety of neutral and charged radicals [4]. These processes illustrate the principle that free radicals produce new free radicals, with the chain only terminated by reactions with another radical or a metal ion. Usually, any H_2_O_2_ can be removed enzymatically. The amounts of energy absorbed by the solution can be readily determined, allowing calculation of the amounts of the primary species generated. When concentrations of the solutes are known, the results can be used to estimate the rates of any competing reactions, because they can provide values of the reaction rate constants, k, in the equation:Reaction rate = k [R^●^] [AH](1)
where the terms in brackets are molar concentrations, R^●^ is a free radical, and AH is a solute.

The wide range of chemical possibilities offered by ionizing radiations are of considerable advantages in the analysis of products, which are typically generated by exposure to ^60^Co gamma rays or to electron beams. They are also particularly useful in disclosing the mechanisms of reactions of radicals. In such studies, the most commonly used technique is pulse radiolysis, when aqueous solutions are exposed to brief, typically ns, pulses of accelerated electrons. The formation and reactions of transient species generated are most commonly followed by changes in absorbance in the μs to ms time scale, although other methods have also been used [1,5,6,7].

### 2.1. Radiation-Generated Free Radicals and Their Targets in Biology

The damaging effects of ionizing radiations on living organisms are well known. The damage is principally oxidative in nature, and not all radiation-induced changes have equal significance for the organism [8]. Only damage to vital molecular targets, generally identified as DNA and lipids, would impair the normal physiological functioning of the organism, while most other targets, such as proteins, would be repaired or replaced, with no detrimental consequences [9]. However, what this last notion ignored was the possibility that attack by the radiation-generated protein radicals, Pr^●^, could propagate the radical chain to other cell components, including the molecules essential for survival [2,10].

Arguably, the most extensively studied of the three main potential targets of ionizing radiations are the polyunsaturated lipids. There are two reasons for this. First, lipid integrity is essential for the maintenance of cell and organelle membranes, preserving their identity and providing both a barrier to the external environment and location for the channels responsible for the selective transport of solutes into and out of cells and compartments. The second reason is the easy oxidizability of unsaturated lipids, especially in condensed phases such as micelles, when a single interaction with a HO^●^ can result in the generation of 100 peroxides [11]. Theoretically, such chains could cause extensive membrane damage from a minor event, as from reaction with a radiation-generated radical. In fact, there is virtually no evidence for the chain oxidation of lipids in biological systems. Rather, attempts to test their possibility have suggested that membrane lipids are not a primary target of radicals generated by radiation. In an important series of experiments, since confirmed, the viability of murine fibroblast cells exposed to X-irradiation was not affected by a fivefold increase in their polyunsaturated lipids content [12,13,14,15]. In other studies, the formation of lipid peroxides in Sp2/0-Ag and U937 cells exposed to gamma radiation showed that lipid oxidation was observed at a barely detectable level, demonstrating effective protection of the membranes by the constituent proteins and other components acting as chain beakers in preventing any extensive lipid oxidation [16].

On DNA, theoretical and experimental evidence shows that while DNA is clearly a vital biological target, it is not the initial site of reaction of the primary HO^●^ generated by radiations. In eukaryotic cells, most of the nucleic acid is in the form of chromatin, where it forms DNA–protein nucleosomes. Numerous studies have shown that the nucleosome DNA is protected from radiation damage by the associated histones and other proteins [17]. Radiolysis of DNA–histone complexes showed that the yields of damaged amino acids were the same as in irradiated separate histones, demonstrating that the proteins were efficient protective agents for the DNA. No significant damage to the DNA bases was observed, even at high radiation doses [18,19]. In the more complex system of T4 bacteriophage, the principal targets of radiation-generated HO^●^ were the proteins, not DNA [20]. The elegant experiments carried out by Ljungman et al. demonstrated by stepwise removal of DNA-bound proteins indicated that they prevented between 93% and 99% of radiation-induced strand breaks in cultured fibroblasts [21,22]. Extensive theoretical and experimental studies by the Spotheim–Maurizot group of the DNA strand breaks induced by HO^●^ identified the presence of histones as the determining factor in DNA protection [23,24,25]. Similar results were reported in several studies using chemically generated radicals [26]. In cultured mouse myeloma Sp2/0 cells exposed to ^60^Co γ rays, DNA fragmentation was evident not immediately after gamma irradiation but only after a considerable delay and showed a pattern of regular chain breaks characteristic of the action of enzymes rather than the random scissions associated with direct effects of DNA radiolysis [16].

The apparent relative immunity of lipids and DNA from direct attack by ionizing radiations or by the radical products of water decomposition leaves proteins as the principal initial site of damage [27]. The chemistry of the effects of ionizing radiation on amino acids and proteins has been investigated for over a century, with the results summarized in numerous publications (reviewed in [1,28,29,30,31]). Knowledge derived from basic research was quickly applied to problems in biology and medicine, prompted by the discovery that living organisms are sensitive to X-rays and nuclear rays [32]. Much of the subsequent work was aimed at identifying the biomolecular targets of ionizing radiations, detecting any induced chemical changes, determining the quantitative relationship between the amount of absorbed energy and any damage, and identifying factors and conditions affecting the extent of the damage. The radical responsible for the damage under the common physiological conditions was identified as the HO^●^, because the e_aq_^−^ are rapidly scavenged by the physiological oxygen [1]. On the basis of abundance alone, with proteins comprising about 70% of the organic mass of cells, they are the most abundant molecules attacked by the HO^●^ [33]. The intracellular molar concentration of proteins is high, between 5 and 10 mM for 50–25 kDa proteins [34], and their rates of reaction with the HO^●^ are typically diffusion-controlled with *k* values some of the highest recorded, typically > 10^10^ M^−1^ s^−1^ [4]. Therefore, there is high probability that in living organisms exposed to ionizing radiations, proteins will suffer a high proportion of the initial damage. While the remaining HO^●^ will react with other biomolecules, the main result is also the formation of aliphatic C-centered radicals, which need to be repaired by the same antioxidants as the proteins damaged in Reaction (2).

### 2.2. Damage Transfer in Proteins

The initial reaction of randomly generated HO^●^ with a protein produces predominantly aliphatic C-centered radicals Pr^●^, close to the site of HO^●^ formation, on amino acid residues located in a solvent-accessible site:PrH + HO^●^ → Pr^●^ + H_2_O(2)
with further Pr^●^ sources from additions of HO^●^ to the aromatic amino acid residues. While the major site of attack in amino acids, peptides, and denatured proteins is commonly the hydrogen of the α-carbon, the reaction in intact globular proteins favors side-chain residues, because of blocking of access by the protein secondary and tertiary structures [31]. Therefore, most of the initial damage will consist of aliphatic C-centered radicals on side chains. There is ample experimental evidence for the transfer of electrons from an initial radical site to other residues, which accounts for the particular sensitivity of Trp, Tyr, His, and Cys in irradiated proteins, although they are not especially prominent occupants of solvent-accessible protein sites [1,6,7,35]. Extensive support for electron transfer from the initially generated amino acid radicals to other residues in peptides and proteins was provided mainly by studies of the oxidation of Trp and Tyr. The HO^●^ generated by pulse radiolysis was converted to less reactive N_3_^●^ or Br_2_^●^^−^, which selectively produced Trp^●^, readily detectable by its absorbance near 510 nm. This gradually decayed to a new peak at 405 nm due to the increased generation of TyrO^●^ by long-range electron transfer (LRET). The LRET was largely intramolecular with first-order kinetics and rate constants in the range of 10^2^ to 10^4^ s^−1^ [1,36,37,38,39,40,41]. Butler et al. [37] pointed out that these findings indicated that the initial site of attack is not necessarily the site responsible for the loss of enzyme activity. A similar mechanism of electron transfer to more electron-deficient sites in proteins exposed to ionizing radiations would explain the selective loss of specific residues and the concept of “radical sinks” [42]. LRET was virtually abolished in the presence of ascorbate with simultaneous formation of the ascorbyl radical [43].

Under normally prevailing physiological conditions, the subsequent reactions of the Pr^●^ are largely determined by the presence of oxygen; the predominant result in its absence is protein aggregation caused principally by radical–radical reactions, while in the presence of oxygen chain, fragmentation predominates, and there is no aggregation [2,9,31]. Therefore, in vivo, the highly probable reaction of the Pr^●^ produces the peroxyl radical, PrOO^●^:Pr^●^ + O_2_ → PrOO^●^(3)

The reaction with these predominantly aliphatic Pr^●^ is fast, even in spite of the typically low 1–50 μM oxygen concentration range in tissues [17], because its rate is diffusion-controlled reactions of C-centered protein radicals, with even the slower reactions of Trp^●^ and Tyr^●^ also producing peroxyl radicals [30,31,33]. Organic peroxyl radicals are good oxidants with reduction potential E^o^’ of 0.77–1.44 V [17], which is able to generate various reactive species, including the hydroperoxides:PrO O^●^ + AH → PrOOH + A^●^(4)
where AH is an electron or H donor [1,33,44]. Both the PrOO^●^ and PrOOH are potentially damaging species that are able to transfer the radical-induced damage to other molecules by direct H abstraction or by generating new radicals and other reactive intermediates [33,45,46], with peroxyl radicals generally considered to be the main chain carriers of radical-induced damage in vivo [47]. In terms of damage to the vital biomolecules, particularly significant reactions of the protein intermediates are with DNA, as demonstrated by the covalent crosslinking of DNA with radiation-generated BSA hydroperoxides and by damage to calf thymus DNA by radicals from histone hydroperoxides [46,48]. Apparently, no comparable studies have been carried out with lipids, but in most of the cell cultures tested, lipid oxidation followed the formation of protein hydroperoxides, suggesting that reactive products derived from decomposition of the PrOOH may have resulted in damage to the lipids [10,16,49]. On the other hand, radical transfer from peroxidizing lipids to proteins has been documented in numerous studies [50].

## 3. Repair of Protein Radicals

### 3.1. Inhibition of Radiation Damage

The discovery of damaging effects of ionizing radiations on living organisms created the need for control of their actions. As a result of its primarily oxidative nature and the common association with free radicals, the obvious damage prevention was by biocompatible antioxidant compounds with low reduction potentials, which are able to donate an electron or H atom to the radical. This would repair the damage, even though often the repair does not always completely restore the damaged molecule, because it can result in changes to its stereochemistry, which is especially significant for proteins, whose biological function requires exact configuration of the amino acid residues in their secondary and tertiary structures.

Based on the knowledge that in biological systems, the main agent of radiation damage is the HO^●^ and its principal targets the proteins, the subsequent most likely fate of the C-centered protein radicals is shown in Figure 1. HO^●^ can also have different sources [2]. However, this is not the topic of the current work.

According to this scheme, the most effective point of repair is reduction of the Pr^●^, preventing its reaction with O_2_ and the generation of a new range of reactive species. The obvious candidates for this role are the endogenous low molecular weight antioxidants, ascorbate (HAsc^−^), urate (H_2_Ur^−^), and glutathione (GSH) [34,43]. This defense system is adequate under normal conditions, but it is insufficient under the unfortunately common condition of oxidative stress, when the oxidative challenge exceeds the antioxidant potential of the organism [51]. Knowledge that oxidative stress, often associated with the formation of excessive amounts of free radicals, causes or aggravates a wide range of human diseases [17] has led to a huge research effort aimed at alleviating the effects of the stress. This should be achievable, in theory, by increasing the effectiveness of the endogenous antioxidants, which is estimated from the knowledge of the concentrations of the individual compounds in vivo and the kinetics of Reaction (5):Pr^●^ + AH → PrH + A^●^.(5)

On concentrations, there is much information on those of HAsc^−^, H_2_Ur, and GSH in various human tissues, but they are tightly controlled, and therefore, their effectiveness cannot be significantly enhanced by oral intake [17,52]. This is also true for the radical scavenging enzymes, which will not be discussed [34]. On kinetics, predictions of the relative abilities of HAsc^−^, H_2_Ur, and GSH to repair protein damage were made difficult by lack of information on the rate constants of Reaction (5), until some recent findings that are summarized in the next section.

### 3.2. Kinetics of Reduction of Pr^●^ by Asc, Urate and GSH

Determination of the rate constants of reduction of protein radicals by these endogenous antioxidants required the application of techniques able to measure the rates of fast reactions. The most commonly used was pulse radiolysis, with optical detection of the formation and decay of free radicals under different energy doses and concentrations of solutes. In most cases, the dissolved proteins were oxidized in solutions by pulses of ionizing radiations in the absence of oxygen. Comparison of the rates of decay of the Pr^●^ in the absence or presence of different concentrations of an antioxidant showed whether it reacted with the Pr^●^, allowing calculation of the rate constant of Reaction (5). For technical reasons, the most commonly measured were the rate constants of reactions of free and protein-bound Trp^●^ and TyrO^●^ with ascorbate, urate, GSH, oxygen, trolox C, superoxide radical, α-Tocopherol, NO, and flavonoids. The *k* values were in the broad range of <10^3^ M^−1^ s^−1^ (TyrO^●^ + O_2_) to 2 × 10^9^ M^−1^ s^−1^ (Trp^●^ + NO) [26,43,53,54]. Ascorbate was selected for many of the kinetic studies; it is the most effective low molecular weight antioxidant in biological systems because it has a low reduction potential of +0.28 V at pH7 [55], is present in most human tissues at significant concentrations [52], and the ascorbyl radical generated in Reaction (5) is almost unreactive. In addition, ascorbate levels decrease in conditions associated with oxidative stress, such as aging, inflammatory diseases, Alzheimer’s and Parkinson’s diseases, and asthma [56,57,58,59,60,61,62]. Comparison of the rates of decay of Trp^●^ and TyrO^●^ in the absence or presence of different concentrations of ascorbate showed whether it increased the rate of the radical decay and allowed calculation of the rate constant of Reaction (6).
Pr^●^ + HAsc^−^ → PrH + Asc^●^^−^(6)

A typical set of measurements of the kinetics of reduction of Trp and Tyr radicals in pepsin is shown in Figure 1.

Similar results were recorded with the other proteins tested: insulin, chymotrypsin, β-lactoglobulin, and lysozyme, with HAsc^-^ accelerating the decay of both the Protein-Trp and -Tyr radicals [64].

Comparisons of the *k* values of the repair reactions showed that only ascorbate might be able to compete with the potentially damaging Reaction (3) and only in the few human tissues with high ascorbate content, such as pituitary and adrenal glands or eye lens [52]. While the concentrations of GSH and urate can be quite high in some human tissues, the relatively low rate constants of their reactions with protein and other C-centered radicals are insufficient for a major general antioxidant role in vivo (Table 1).

An unexpected feature of the results listed in Table 1 is the close similarity of the rate constants of reactions of the free and protein-bound Trp radicals, indicating the absence of any significant barrier to the access of the antioxidants imposed by the lysozyme and several other proteins [43]. Examples of the rate constants of reactions of C-centered amino acid and protein radicals with the endogenous antioxidants are listed in Table 2. The highest values are for the reaction of Ac-Trp^●^ amide, which is a good model for a Trp residue in proteins, either in free solution or bound in proteins, except for a lower constant for β-Lactoglobulin-Trp^●^ [64]. The corresponding *k* values for the TyrO^●^ in proteins are more than an order of magnitude lower than the much higher value for the free Ac-TyrO^●^ amide, and again a low value for the β-Lactoglobulin-TyrO^●^, which is probably due to the protein structure restricting access to the HAsc^-^.

Comparison of the rate constants of reactions of ascorbate with free and protein-Tyr and -Trp radicals provided new insights into the general ability of antioxidants to repair these radicals in proteins. However, as demonstrated above, C-centered radicals produced by the radiation-generated HO^●^ will not be confined to the Trp and Tyr residues, but rather, they involve any amino acid encountered by the HO^●^. Investigation of the kinetics of the reduction by antioxidants of such randomly produced radicals was so far reported out only once, which was probably because of the expectation that a single rate constant would only be possible if all protein radicals reacted with ascorbate at the same rate or only one kind of radical was produced. Both cases seemed unlikely. However, tests with five proteins showed that in the case of ascorbate, *k* values for Reaction (6) could be derived [65]. The kinetics of formation of the Asc^●^^−^ could not be analyzed by a single exponential but required two, showing the occurrence of two simultaneous reactions: a fast one, accounting for 10–20% of the total Pr^●^ formed in Reaction (2), and a slower one accounting for the rest (Figure 2).

The fast phase (Figure 2B) was interpreted as reflecting the reaction of ascorbate with radicals generated on amino acid residues Cys, Trp, and Tyr in the protein surface, whose reaction rate constants are between 1 × 10^7^ M^−1^ s^−1^ and 1.2 × 10^9^ M^−1^ s^−1^ [64,66]. The slow phase was interpreted as reflecting the reaction of HAsc^-^ with Cys, Trp, and Tyr radicals generated by LRET from the surface amino acid radicals reacting slowly with HAsc^−^ [65]. As a unimolecular process, with a typical value of k = 2 × 10^2^ s^−1^ for native lysozyme at pH 7, LRET can compete with the bimolecular Reaction (5) because of the low reactant concentrations. The maximum *k* value of the slower reaction phase, ≈ 2 × 10^7^ M^−1^ s^−1^, is similar to that of HAsc^-^ and protein-TyrO^●^ derived for most of the proteins tested (Table 2). These results allow the calculation of an approximate distribution of Pr^●^ in human tissues between reactions with oxygen or ascorbate (Figure 1, with AH = HAsc^−^).

### 3.3. Fast Inactivation of C-Centered Radicals by Aromatic Compounds

The principal causes of the inability of ascorbate and the other endogenous antioxidants to prevent the occurrence of radical chain reactions under physiological conditions are the relatively low rate constants of their reactions with Pr^●^ (Table 2) and insufficient tissue concentrations. On the kinetic requirement, according to the classical definition, oxidation consists of the “complete removal of one or more electrons from a molecule” [71]. In most cases of phenols and polyphenols, a reaction such as (5) is slow because of an entropic barrier caused by exchange of at least two protons and breaking of the O-H bond [72]. However, recent studies of the mechanism of reactions of a range of radicals with aromatic compounds demonstrated that the electron transfer in the classical definition of oxidation is only a final slow step, which is preceded by a reversible fast formation of an adduct radical. Adduct formation is followed by other equilibria before the final electron transfer step, with each stage in this chain producing a radical with lowered reactivity, as required by thermodynamics. Based on the observation of fast adduct formation between aromatic molecules and C-centered radicals generated by radiation in alcohols and carbon halogen compounds, it seemed likely that similar adducts might form with aliphatic peptide C-centered radicals and the aromatic polyphenols (PP-OH) [73]. The proposition was tested by pulse radiolysis of solutions of containing a polyphenol and a 200-fold higher concentration of an amino acid. Most of the experiments involved the flavonol morin, but similar results were recorded with rutin, gallate, and epigallocatechin gallate. With morin, pulse radiolysis resulted in fast absorbance growth at 375 nm characteristic of the morin radical [73]. The half-life of the absorbance increase was less than 1 μs, giving the bimolecular reaction rate constant of 1 × 10^10^ M^−1^ s^−1^ (Figure 3).

Subsequently, as shown in Figure 4 (red line), there was a brief decay of the absorbance (stage II), which was followed by slower growth to a more stable maximum (stage III). Radiolysis of morin alone produced morin radicals, which decayed in a bimolecular radical recombination (blue line).

In addition to the C-centered radicals *N-*Ac-Ala^●^-NH_2_, similar fast adduct formation was recorded with morin and with the radicals of the acetyl derivatives of Lys, Pro, and Gly, as well as of Cyclo(Gly)_2_ and human serum albumin. The rate constants of the adduct formation were close to diffusion-controlled with values of (1–10) × 10^9^ M^−1^ s^−1^ [73]. We interpreted the mechanism of the interaction as consisting of a fast initial reversible formation of the C-centered radical-polyphenol adduct, which was followed by subsequent equilibria, leading finally to the slower endothermic classical Reaction (2):(Peptide)^●^ + PP-OH **⇌** (Adduct_1_^●^) **⇌ ⇌ ⇌** (Adduct_n_^●^) → Peptide + PPO^●^.(7)

### 3.4. Polyphenols and Their Metabolites as Biological Antioxidants: New Insights

Discoveries of the damaging actions of ionizing radiations in humans, and of the inability of the endogenous antioxidants to protect them under conditions of oxidative stress, has produced thousands of attempts to find agents capable of controlling the damage. Given the importance of the reactions outlined in Scheme I, the agents should be capable of repairing protein C-centered radicals, biocompatible, able to compete in vivo with Reaction (3), and easy to administer, preferably by oral intake. As outlined in Section 3.3, a major group of compounds potentially capable of fulfilling these requirements is the dietary flavonoids and other polyphenols.

Polyphenols form a diverse group of about 8000 secondary plant metabolites, with over 500 forming part of the human diet [74,75]. Many studies have demonstrated the beneficial effects of diets rich in polyphenols in conditions such as mortality, cancer, atherosclerosis, cardiovascular disease, diabetes, inflammation, ischemia, hypertension, stroke, metabolic syndrome, obesity, urinary tract infections, and conditions associated with aging, such as cognitive degeneration and Alzheimer’s and Parkinson’s diseases. [76,77,78,79,80,81,82,83,84]. An early theory explaining the health benefits was based on the radical scavenging properties of polyphenols, because many diseases and other deleterious health conditions are apparently associated with the excessive formation of free radicals. However, the antioxidant theory has been questioned and virtually abandoned as the result of three sets of experimental observations: (1) slowness of the polyphenol–radical reactions, (2) loss of phenolic hydroxy groups through metabolism, and (3) low concentrations of the polyphenols in tissues [74,83,85,86].

The virtual abandonment of any significant antioxidant role of polyphenols in vivo is now challenged by experimental results indicating that it is based on incomplete knowledge of the mechanism of the polyphenol-free radical reaction. The classical mechanism assumes that the antioxidant action required transfer of electron from the polyphenol (PP-OH) to the radical:PP-OH + R^●^ → PPO^●^ + RH.(8)

In contrast, new results demonstrate that (1) the polyphenol adduct formation is very fast and lowers the activity of the C-centered radicals; (2) loss of phenolic groups does not abolish the antioxidant ability of the polyphenols; and (3) the antioxidant defenses of an organism include polyphenols and their aromatic metabolites, which are present in vivo at effective concentrations.

Formation of Adduct ^●^ (Reaction 7) is close to diffusion-controlled with rate constants of 10^9^–10^10^ M^−1^ s^−1^ [73]. It involves no breaking of bonds and is already an antioxidant step, requiring only a C-centered radical and an aromatic group in the reacting entities but not the phenolic hydroxy groups [73,87,88,89,90,91]. Based on the role of the aromatic groups, the human antioxidant arsenal would include not only the parent polyphenols but most of their metabolites retaining the aromatic character [75]. An additional major source of polyphenol-derived antioxidants is the easily absorbed metabolites of the high molecular weight condensed non-extractable polyphenols. These are catabolized by gut microorganisms, can reach >10 μM concentrations in tissues, and have antioxidant, anti-inflammatory, and antiatherogenic properties [92,93,94,95,96,97,98]. The resultant new estimates of the plausible concentrations of polyphenols and their aromatic metabolites in vivo allow an approximate evaluation of their ability to scavenge C-centered protein radicals in competition with oxygen. Assuming the upper limit of [O_2_] at 13 μM in cells [17] and rate constants of adduct formation between Pr^●^ and polyphenols and their aromatic metabolites of 5 × 10^9^ M^−1^ s^−1^ [73], with k = 2.4 × 10^9^ M^−1^ s^−1^ for the reaction (amino acid)^●^ + O_2_ [99], 50% scavenging of the Protein^●^ in competition with Reaction (3) requires a 6.2 μM concentration of the aromatic antioxidants. Such levels of sustained concentrations should be easily achievable in vivo from a diet rich in vegetable content and may be sufficient to tip the balance from oxidative stress to antioxidant protection [75,92,93].

## 4. Conclusions

The oxidation of proteins in individuals exposed to ionizing radiations is a major hazard to health. Since the normal endogenous antioxidant systems cannot repair damage to cells and tissues under oxidative stress, additional defenses are required. Recent experimental results, based on pulse radiolysis measurements, show that flavonoids, polyphenols, and their aromatic derivatives may be good candidates for this role. Since this proposition is so far based on results obtained in vitro with pure compounds, much new research needs to be conducted with more complex biological systems to determine if increased radiation protection can be achieved by an increased presence of biocompatible aromatic compounds. If this proves to be the case, new drugs could be developed for administration to individuals subjected to transient of chronic oxidative stress because of their occupation or location.

## Data Availability

All data presented in this manuscript.

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
