# Peer review of "Initiation and Prevention of Biological Damage by Radiation-Generated Protein Radicals"

_ijms, 2021, doi:10.3390/ijms23010396_

Round 1

Reviewer 1 Report

This review by Gebicki and Nauser is an interesting piece of work describing the mechanisms of production and secondary reactions of protein radicals generated by ionizing radiations. The work is well presented, easy to follow and informative. Particular emphasis is given to the potential role of polyphenols as repairing agents of protein damage. I recommend its publication in the International Journal of Molecular Sciences after consideration of some (minor) points:

1.- Section 2.2

I understand this section describes damage transfer in proteins. However, I would suggest to mention some termination reactions, for example, those mediated by radical-radical reactions. Evidence has shown self-reactions of Trp· and Tyr· to generate di-Trp and di-Tyr crosslinks, respectively, with kinetics constants ~ 2-6 x 108 M-1s-1 (Carroll, L. et. al. Free Radic Biol Med 2017, 113, 132-142). Protein structure and dynamics limit this kind of reactions, however, both species are relevant in biological/pathological contexts (i.e. Coelho et. al. J Biol Chem 2014, 289, 30690-30701).

Reaction (3). I agree this is diffusion controlled. However, if Pr· is formed at the side chain of Trp (indole ring) or Tyr (phenol ring), kinetics of the reaction with O2 are modest (k < 1 x 103 M-1 s-1 for Tyr· and k £ 4 x 106 M-1s-1 for Trp·, Houée-Lévin, C. et. al. Free Radical Res 2015, 49, 347-373, ), modulating the pathways of Pr· decay. In spite of this, peroxyl radicals derived from Trp and Tyr are produced in proteins (refs, 30,31 and 33).

2.- Section 3.4 (bottom), “…in competition with Reaction (3) requires a 6.2 μM concentration of the aromatic antioxidants. Such levels of sustained concentrations should be easily achievable in vivo from a diet rich in vegetable content and may…”

Please include a reference supporting that micromolar concentrations of polyphenols can be reached after intake of vegetable rich diets. An interesting point (not mentioned by authors) is the possible formation of protein radicals in gastro-intestinal (GI) tissue. High oxidative stress levels are associated with pathological conditions (i.e. inflammatory bowel disease) and high polyphenolic concentrations are easily reached in GI tract. 

3.- Format of tables and references should be checked.

Reviewer 2 Report

The review article “Initiation and prevention of biological damage byradiation-generated protein radicals” by Janusz M. Gebicki and Thomas Nauser aims to document the progress in the field of radiation generated protein damage and its prevention using small molecules. It is timely review considering the demand for protein based biological molecules used as therapeutics. The authors have reviewed vast majority of the literature and clearly presented it.

I am enthusiastic about this review article and supportive of its publication. I only offer some minor suggestions to improve readability and enhance the message of the paper (adopting them is optional).

Minor issues:

  1. There is spacing mistake in title.
  2. Authors have addressed the issue of protein damage by irradiation. Since it’s a review article, it would be good if authors could provide 1-2 illustrations to demonstrate the problem and how use of small molecules could avoid it.
  3. It will be good to provide context of protein-based therapeutics as it is not just a problem within cell bur also in storage vials.
  4. What excipients are currently in use if any to avoid such irradiation related issues?
  5. Scientist now use Small-angle X-Ray scattering to understand the radiation damage issues. Authors should highlight SAXS in their review.
  6. Authors should also mention negative aspects of use of polyphenols.
  7. Are there any FDA approved molecule that could substitute the use of molecules mentioned in here by authors?
  8. Reference 23-25 format should be corrected.
